# LD-SDM: Language-Driven Hierarchical Species Distribution Modeling

## Abstract

We focus on species distribution modeling using global-scale presence-only data, leveraging geographical and environmental features to map species ranges, as in previous studies. However, we innovate by integrating taxonomic classification into our approach. Specifically, we propose using a large language model to extract a latent representation of the taxonomic classification from a textual prompt. This allows us to map the range of any taxonomic rank, including unseen species, without additional supervision. We also present a new proximity-aware evaluation metric, suitable for evaluating species distribution models, which addresses critical shortcomings of traditional metrics. We evaluated our model for species range prediction, zero-shot prediction, and geo-feature regression and found that it outperforms several state-of-the-art models. We will share code, data, and model checkpoints after acceptance.

## 1 Introduction

Species distribution modeling (SDM) is a challenging remote-sensing task that involves establishing the relationship between geographical, environmental, and biophysical conditions and species presence. The goal of SDM is to produce large-scale range maps for species. Such maps can then be used to develop tools for modeling habitat suitability, predicting future distributions, etc. These tools are essential for biodiversity conservation and understanding the environmental effects of climate change (Beery et al., 2021).

Traditionally, species ranges were mapped using geographical and environmental features. The approaches were either limited to single-species models or modeled a fixed set of species. While the current approaches have their merits, we contend they fall short in accurately representing the inter-species relationships inherent in the taxonomic hierarchy. For example, *species* belonging to the same *genera* tend to be found in similar locations. They often share many common characteristics, including their general habitat preferences (Thorson et al., 2016). This pattern diminishes as one moves up in the taxonomic hierarchy. To capture these complex hierarchical relationships between the species, we use large language models (LLMs) to encode their taxonomic information in text (as shown in Figure 1) in the form of species-specific embeddings. Moreover, by leveraging the vast knowledge of LLMs, our approach enhances the identification of rare taxa (Caradima et al., 2019). To our knowledge, no previous work has incorporated LLMs for SDM.

Previous approaches to SDM (Mac Aodha et al., 2019; Cole et al., 2023; Lange et al., 2023) tried to answer the question, "Which species are likely to be found at a given location?" However, to incorporate species-specific rich information into SDM, it is necessary to reformulate the SDM problem. Therefore, in this paper, we reformulate the SDM problem as "Which locations are likely for a given species to be found?" Assuming uniform prior over the species distribution (Mac Aodha et al., 2019; Lange et al., 2023), this is an equivalent formulation. But by doing so, we can now incorporate a variety of information, including species-specific embeddings or any other metadata, which opens up new opportunities for more accurate modeling. Alternatively, this formulation makes it easier to incorporate unseen species that were not present in the training data.

Previously, SDMs were trained using environmental features while ignoring the geographic location aspect. Recently, several works showed the benefit of incorporating geographic locations in various applications including SDM (Cole et al., 2023; Botella et al., 2018; Tang et al., 2018) and fine-grained image classification (Ayush et al., 2021; Mac Aodha et al., 2019; Chu et al., 2019). A series

“{**class**: *Aves*, **order**: *Charadriiformes*}”

(a)

“{**class**: *Aves*, **order**: *Charadriiformes*, **family**: *Scolopacidae*}”

(b)

“{**class**: *Aves*, **order**: *Charadriiformes*, **family**: *Scolopacidae*, **genus**: *Calidris*}”

(c)

“{**class**: *Aves*, **order**: *Charadriiformes*, **family**: *Scolopacidae*, **genus**: *Calidris*, **species**: *Calidris alba*}”

(d)

Figure 1: **Species Distribution Modeling using LLM Guidance.** We use text prompts in the form of key-value pairs for training our species distribution model. This kind of prompt captures the taxonomical hierarchy of species. We use only the exhaustive prompt (d) for training while the rest are used for zero-shot predictions.

of works (Bonev et al., 2023; Rußwurm et al., 2023) proposed spherical harmonics representation of geographic location. Inspired by these works, we propose to use the Spherical Fourier Neural Operators (SFNO's) (Bonev et al., 2023) as a geographical and environmental feature extractor for SDM.

One of the principal challenges in training SDM models with presence-only data is properly evaluating the resulting species distributions. One approach is to compare to hand-crafted range maps, but such maps are difficult to create and not widely available with sufficient quality to be suitable. To this end, we propose a proximity-adjusted evaluation metric that penalizes predictions based on their distance to the closest true locations. We construct a One-Way Probability-Weighted Chamfer Distance (PWCD) metric, that measures the dissimilarity between the true and predicted species range maps, which is robust to small shifts. This enables an effective evaluation of SDMs using pixel-level range maps and crowdsourced data (where absence is not confirmed).

The key contribution of our work are as follows:

1. We use large language models to encode taxonomic information of species, which allows mapping over any taxonomic level.

2. We reformulate the SDM problem which enables the use of species-specific embeddings. This allows inference on unseen species.

3. We propose a novel proximity-aware metric for assessing the performance of SDMs using pixel-level range maps.

## 2 RELATED WORKS

Species distribution modeling (SDM) methods use environmental, geographic, and species observation data to predict species distribution across geographic space and time (Elith & Leathwick, 2009; Beery et al., 2021). In this context, we restrict ourselves to species range mapping over geographic space only. In the past, SDM relied on hand-crafted geographic and environmental features as input to capture the implicit spatial correlations in data (Elith* et al., 2006; Norberg et al., 2019; Valavi et al., 2022). These approaches were limited to one species per model due to their reliance on traditional supervised learning algorithms. With advancements in deep neural network architectures, end-to-end frameworks can model joint species distributions and complex implicit relations in data (Botella et al., 2018; Chen & Gomes, 2019; Cole et al., 2023).

Another important aspect of SDM is the type of species occurrence data. Species occurrence data may exist in one of the two forms: *presence-only* and *presence-absence*. Collecting presence-absence observations is challenging, requiring expert surveys especially to confirm the absence of a given species in some region (MacKENZIE, 2005). However, once collected, this kind of data can be easily fed into supervised learning frameworks, without requiring sophisticated modifications (Fink et al., 2010; Chen et al., 2016; Franceschini et al., 2018; Botella et al., 2018; Teng et al., 2023a). With no expert knowledge, presence-only data can be recorded when any species of interest is encountered. This way of collecting data does not require the data collector to confirm absences. However, since there are no negative samples, it is difficult to use standard supervised learning frameworks (Chen & Gomes, 2019; Johnston et al., 2020; Botella et al., 2021).

Most recent works formulate the problem of SDM using presence-only data as a multi-label classification task (Mac Aodha et al., 2019; Cole et al., 2023). Here, the goal of SDM is to predict the likelihood of species for a given location and environmental features. Many multi-label learning losses have been proposed in the literature that can easily be plugged in for this task. For instance, the AN-full (full assume negative) loss (Mac Aodha et al., 2019) randomly samples locations that are assumed to be negative during training. The ME-full (full maximum entropy) (Zhou et al., 2022) tries to maximize the entropy of predictions of negative samples and at randomly sampled locations during training. Other losses such as ASL (asymmetric loss) (Ridnik et al., 2021), Hill loss (Zhang et al., 2021), and RAL (robust asymmetric loss) (Park et al., 2023) weigh the negative samples in a way that easy negative and potential false negative samples are down-weighted. Cole et al. (2023) evaluated the performance of SDMs trained with losses involving pseudo-negative sampling. We extend their work by evaluating the impact of training their model with the ASL and RAL losses.

Several studies have attempted to assess the performance of species distribution models trained on presence-only data (Segurado & Araujo, 2004; Wilson, 2011; Norberg et al., 2019; Konowalik & Nosol, 2021; Valavi et al., 2022). They measure performance using Kappa index, Kullback–Leibler divergence, area under the curve (AUC), or mean average precision (MAP). However, these metrics are inadequate in quantifying the performance of models on pixel-level data. It is worth noting that none of these metrics consider the spatial proximity of predictions to their ground-truth observations, hence making them sensitive to spatial shifts in predictions.

Finally, another line of work uses high-resolution satellite imagery at each species occurrence location to guide SDM methods (Botella et al., 2019; Cole et al., 2020; Lorieul et al., 2021; 2022; Sastry et al., 2024; Teng et al., 2023b; Dollinger et al., 2024). For example, (Teng et al., 2023b) uses satellite imagery and geolocation to directly predict the likelihood over species while (Sastry et al., 2024) uses a contrastive learning framework to establish similarity between ground-level images of species and corresponding satellite imagery. These methods are limited in geographic scale due to the high-resolution nature of raw data (Cole et al., 2023). Except Teng et al. (2023b); Dollinger et al. (2024), all the works view this problem as an information retrieval task rather than a species prediction task.

## 3 METHOD

### 3.1 PROBLEM DEFINITION

We consider presence-only data in the form of occurrence records: $\mathcal{D} = \{(l_i, y_i)| \ i = 1, ..., N\}$, where $l_i \in \mathbb{R}^2$ is the geolocation of each observation and $y_i \in [1, ..., S]$ is the species of the observed organism. Additionally, a set of environmental features $e_i \in \mathbb{R}^n$ is included for each location $l_i$. Many previous studies (Mac Aodha et al., 2019; Lange et al., 2023; Cole et al., 2023) have tackled the task of SDM by training a neural network to model the conditional probability of $p(y| l, e)$. This translates to the problem of "How likely a *species* can be found at a given *location*?" However, our approach involves modeling the probability of $p(r| y, l_r, e_r)$, which answers the question "Is it likely for a given species to be found at a given location?" Here, $r \in [1, ..., L]$ is a set of discrete locations that are of interest when predicting the range of species belonging to some class. In this framework, we define the set of locations $r$ as a contiguous grid that is laid over the surface of the globe. The variables $l_r$ and $e_r$ represent fixed parameters in the model, namely the location vectors and environmental features defined at each location in $r$ respectively. It is easy to see that the input $y$ and the parameters $(l_r, e_r)$ can have any arbitrary representations. So, this new formulation allows for flexible encoding of both the species class and geographic location, compared to the previous formulation which only allowed flexibility in representing the geographic location.

### 3.2 ARCHITECTURE OVERVIEW

Our objective is to train a neural network of the form: $f_\theta(y; l_r, e_r) : \mathbb{R}^d \rightarrow [0, 1]^L$, where $d$ is the dimensionality of the species feature vector. $f_\theta$ is composed of three modules: 1) a location encoder network $f_\alpha$, 2) a species encoder network $g_\beta$, and 3) a multi-label classifier $h_\gamma$. See Figure 2 for a high-level architecture diagram. The location encoder network $f_\alpha$ consists of convolutional feature extractors and the Spherical Fourier Neural Operator (SFNO) (Bonev et al., 2023) blocks that take as input the fixed parameters: $l_r$ and $e_r$. The network processes the global environmental covariates

Figure 2: **Proposed Architecture**. During training, features extracted from the environmental covariates are added to a learnable positional embedding and passed to the spherical Fourier neural network (SFNO). The species range map is obtained by computing cross-attention between the representations from SFNO and embeddings obtained from LLaMA-2 for a text description of a given species. Inference requires only cross-attention computation if embeddings are precomputed.

$(e_r)$ to extract feature maps that represent the geography of a region. To incorporate geographic location-based features, we use sin-cos location encodings $(l_r)$ as learnable positional encodings for the SFNO blocks. The species encoder $g_\beta$ is a large language model that takes as input the text descriptions corresponding to a particular species of class $y$. This allows for a flexible representation of species-specific features in the form of natural language. The multi-label classifier $h_\gamma$ consists of a cross-attention network that fuses features from the SFNO and text encoder and performs multi-label classification using a linear layer. We call our model Language-Driven Species Distribution Model (LD-SDM).

### 3.3 SPECIES-SPECIFIC EMBEDDINGS

One of the novel contributions of this paper is incorporating natural language descriptions of species for SDM. Many popular crowdsourcing science platforms, such as iNaturalist (Van Horn et al., 2018), contain not just occurrence records, but also valuable species-specific information. This information may include the complete taxonomic hierarchy of each species (such as class, order, family, etc.) or the IUCN red list category (such as endangered, near threatened, etc.). While often ignored, this fine-grained information could be incorporated to enhance the performance and interpretability of SDMs. In this work, we incorporate the taxonomic hierarchy of species into our SDM in the form of text. Each species is uniquely identified by its taxonomic classification ranks, which can be represented as key-value pairs. For instance, the key-value pairs that represent the *wild duck* are defined as: t = {**class**: *Aves*, **order**: *Anseriformes*, **family**: *Anatidae*, **genus**: *Anas*, **species**: *Anas platyrhynchos*}.

Recently, Large Language Models (LLMs) have demonstrated excellent capabilities in representing implicit knowledge, linguistic structure, and hierarchical concepts when trained on massive amounts of data (Anil et al., 2023). With no finetuning, pretrained LLMs have shown impressive zero-shot performance with zero in-domain data. In this work, we use the recently released, LLaMA-2 (Touvron et al., 2023) to encode the species-specific key-value pairs as text. We experiment with three variants of LLaMA-2 of which the 70B model produced the best performance.

### 3.4 LOSS FUNCTIONS

Species distribution modeling is a special case of multi-label learning. Numerous losses for multi-label learning have been evaluated for SDM. Depending on the formulation of the SDM task, it may fall in the domain of Single Positive Multi-Label Learning (SPML), where only a single presence of species is known for a specific geographic location. We describe the loss functions considered in this work below and provide extensive experimentation to assess their impact in Section 4.4.

**Full Assume-Negative (AN-Full).** This loss is a kind of pseudo-negative sampling multi-label learning loss based on the argument that the majority of the species are absent at any given location (Cole et al., 2023). This is typically used in SPML problems. The loss is given by:

$$\mathcal{L}_{\text{AN-full}}(\hat{y}, y) = -\frac{1}{S} \sum_{s=1}^{S} [\mathbb{1}_{[y_s=1]} \lambda log(\hat{y}_s) \tag{1}$$
$$+ \mathbb{1}_{[y_s=0]} log(1 - \hat{y}_s) + log(1 - \hat{r}_s)]$$

where $\hat{r}$ is a uniformly randomly sampled location from the spatial domain of the globe.

**Full Maximum-Entropy (ME-Full).** This loss was introduced in (Zhou et al., 2022) to solve SPML problems. The idea of this loss is to maximize the entropy of predictions for negative samples and for randomly sampled pseudo-negative samples. The loss is given by:

$$\mathcal{L}_{\text{ME-full}}(\hat{y}, y) = -\frac{1}{S} \sum_{s=1}^{S} [\mathbb{1}_{[y_s=1]} \lambda log(\hat{y}_s) \tag{2}$$
$$+ \mathbb{1}_{[y_s=0]} H(\hat{y}_s) + H(\hat{r}_s)]$$
$$H(\hat{y}) = -[\hat{y}log(\hat{y}) + (1 - \hat{y})log(1 - \hat{y})]$$

where again $\hat{r}$ is a uniformly randomly sampled location.

**Asymmetric Loss (ASL).** This loss, proposed by (Ridnik et al., 2021), weights positive and negative samples differently. The positive and hard negative samples are given more weight. It is a fairly general loss for multi-label learning which can also be used for SPML, without the need for psuedo-negative sampling. This loss is given by:

$$\mathcal{L}_{\text{ASL}}(\hat{y}, y) = -\frac{1}{S} \sum_{s=1}^{S} [\mathbb{1}_{[y_s=1]} (1 - \hat{y}_s)^{\gamma^+} log(\hat{y}_s) \tag{3}$$
$$+ \mathbb{1}_{[y_s=0]} (p_s^\tau)^{\gamma^-} log(1 - p_s^\tau)]$$

where $p^\tau = \max(\hat{y} - \tau)$ is used for hard thresholding easy negatives.

**Robust Asymmetric Loss (RAL).** Recently, (Park et al., 2023) proposed an improved version of the ASL loss by combining the Asymmetric Polynomial Loss (APL) (Huang et al., 2023) and the Hill loss (Zhang et al., 2021), which makes it less sensitive to hyperparameters. This loss is given by:

$$\mathcal{L}_{\text{RAL}}(\hat{y}_s, y_s) = [\sum_{m=1}^{M} \mathbb{1}_{[y_s=1]} \alpha_m (1 - \hat{y}_s)^{m+\gamma^+} \tag{4}$$
$$+ \sum_{n=1}^{N} \mathbb{1}_{[y_s=0]} \psi(\hat{y}_s) \beta_n (p_s^\tau)^{n+\gamma^-}]$$

where $\psi(\hat{y}_s) = \delta - \hat{y}_s$ is the Hill loss term. $M$, $N$, $\alpha_n$ and $\beta_n$ are balance parameters for positive and negative samples.

## 4 EXPERIMENTS

In this section, we describe the implementation details and performance of our models compared to the state-of-the-art models on the task of species range prediction and geo-feature regression. Additional quantitative results are reported in the appendix.

### 4.1 IMPLEMENTATION DETAILS

As described in Section 3.2, we use SFNO (Bonev et al., 2023) to extract geographic and environmental-specific features. We use two spectral and encoder layers in the SFNO. The encoder

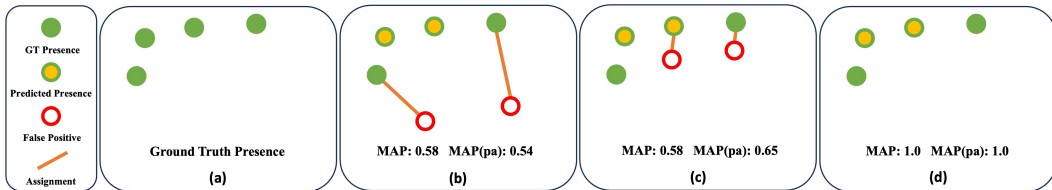

Figure 3: Performance of our spatially-aware metric against the MAP score. The figure depicts the inability of the MAP score to distinguish between range maps with varying spatial distributions of false positives. (a) Ground truth range map. (b) Predicted range map with false positives spatially distant from true positives. (c) Predicted range map with false positives spatially close to true positives. (d) Predicted range map without false positives.

dimension used is 128 while the number of SFNO blocks are two. For the coordinate-only models, we drop the environmental features during the training. For the text encoder, we use *frozen* LLaMA-2 (Touvron et al., 2023) (70B variant) and extract its encoder outputs for each species. Before training, we precompute and save the embeddings for each species in the disk. For the multi-label classifier, we use a single-layer multi-headed cross-attention network followed by a fully-connected layer with sigmoid activations. We compare LD-SDM with two state-of-art-models: SINR (Cole et al., 2023) and SIREN(SH) (Rußwurm et al., 2023). SINR learns an end-to-end neural representation of geographic locations that is used to predict the likelihood over species. SIREN(SH) uses a spherical harmonic representation of geographic locations which is then used to predict the likelihood over species. The rest of the details about our models and the implementation of state-of-the-art models are described in Appendix C.

## 4.2 PROXIMITY-AWARE EVALUATION

When predicting species range, the evaluation process typically involves using range maps defined by experts or curated carefully (Norberg et al., 2019; Valavi et al., 2022). However, these maps can often be sparse and difficult to obtain. Alternatively, using crowdsourced species observation data is challenging since the absence of a given species is not recorded. Intuitively, models with false predictions close to their true locations are better than those with distant false predictions. This is because models in the former case learn *spatially localized* species distributions.

Previous studies have used the area under the curve (AUC) (Norberg et al., 2019; Valavi et al., 2022) and the mean average precision (MAP) (Cole et al., 2023) to evaluate SDMs. However, these metrics ignore the spatial distribution of predictions and weigh them equally, making them highly sensitive to spatial shifts in the predictions. This makes evaluating SDMs a challenging task since it requires the comparison of spatially distributed probability values. Pixel-wise comparison methods are not well-suited for this task since they are susceptible to spatial prediction shifts.

A well-known spatially explicit distance used for comparing any two distributions is the Earth-Mover (Wasserstein) Distance. Due to its computational requirements and infeasibility on large-scale data points, Chamfer Distance is a popularly used proxy (Bakshi et al., 2023). However, it has only been developed for point clouds and there doesn't exist a chamfer distance measure to compare two spatial probability distributions. Inspired by (Wu et al., 2021), we modify their Density-Aware Chamfer Distance (DCD) metric that works for comparing a *predicted* probability distribution map ($P_1$) with a *binary* probability map ($P_2$) representing the ground truth. We add a likelihood probability term which weights the distance term. We call this metric One-Way Probability Weighted Chamfer Distance (PWCD), which measures the false positive rate in a proximity-aware manner. This metric is between [0,1] and could also be used to compute the number of false positives instead of the rate. It is given by:

$$d_{\text{PWCD}}(P_1, P_2) = \frac{1}{|N|} \sum_{x \in P_1} \left(1 - e^{-\alpha p(x)||x-y||_2}\right) \tag{5}$$

$$FP_{\text{PWCD}}(P_1, P_2) = \sum_{x \in P_1} \left(1 - e^{-\alpha p(x)||x-y||_2}\right) \tag{6}$$

where, $y = \underset{z \in P_2 \cap p(z)=1}{argmin} ||x-z||_2$. This represents the nearest neighbor ground-truth observation of a given pixel $x$ in the predicted probability distribution map. $N = \{z \in P_2 | p(z) = 0\}$ is the set of all negative observations in the ground truth. $\alpha$ is a temperature parameter that controls the exponential weighting of the distance. See Appendix B for more details. For our case, $|P_1| = |P_2| = L$. Notice that we have omitted the density term in the metric as our probability maps are a contiguous grid of pixels and equally dense everywhere. Since it is a proxy for false positives, it can be plugged into any other metric of choice such as the MAP and AUC. In this work, we report proximity-adjusted MAP scores denoted as MAP(pa) by replacing false positives with our metric. All the scores reported are an average over scores computed for each species category. Figure 6 illustrates three different scenarios of predicted species range maps. While the MAP score yields identical results for cases (a) and (b) despite the spatial distribution of the predictions being different, our proposed metric penalizes distant false predictions and is capable of evaluating the predictions in both cases more accurately. However, in case (c) where there are no false predictions, both MAP and MAP(pa) produce the same results.

### 4.3 TRAINING AND EVALUATION

We focus on bird species occurrence data which is abundantly available yet challenging to model as compared to other species due to complex migration patterns. In Table 1, we describe the overview of each dataset compiled in this work. We compiled 131.7 million observations of 4141 species of birds using the GBIF (`gbif.org`) platform which includes data from various community science platforms. This data is collected for the years 2012-2020. For evaluation, we use curated research-grade observations from the iNaturalist (Van Horn et al., 2018) and eBird (Sullivan et al., 2009) for 2021 and 2022. These contain 17.9 million and 17.2 million observations respectively. Using the same platform, we created a dataset of 84 species that are unseen (not present in training data) and rare ($< 1000$ observations) for zero-shot performance evaluation. This contains a total of 26k observations. These platforms are useful since misclassification errors are highly mitigated due to community consensus (Lange et al., 2023), hence being a strong representative of the ground truth. Further, we use the same procedure as described in (Cole et al., 2023) for the geo-feature regression task.

Table 1: Number of observations in each dataset compiled in this work.

| Dataset | Year | #Observations | #species |
|---|---|---|---|
| Train | 2012-2020 | 131.7M | 4141 |
| Test | 2021 | 17.9M | 3960 |
| Test | 2022 | 17.2M | 4010 |
| Zero-Shot | 2021-2022 | 26k | 84 |

All the training and evaluations are done at a spatial resolution of $0.2^o$. This implies $L = 1,620,000$. During training, we create ground-truth pixel-level range maps for each species at $0.2^o$ resolution using the observations present in the occurrence data. For this, we use the histogram binning technique and clip all the values greater than one to one (Cole et al., 2023). These maps are then used to calculate the various multi-label learning losses described previously. Finally, we report MAP(pa) and $d_{PWCD}$ using $\alpha = 0.1$ for the species range prediction task and $R^2$ for the geo-feature regression task. See Appendix C for more details on the training settings and hyperparameters used.

### 4.4 RESULTS

**Quantitative Evaluation.** In Table 2, we report the performance of all the models considered in this work. LD-SDM has shown better performance than the other models on the task of species range prediction. The zero-shot performance of LD-SDM on unseen species is reasonably good. Although it has an increased false-positive rate, it can be attributed to the model using higher-level taxonomic relationships. Our metric is sensitive to model output confidence ranges. SINR and SIREN(SH) output likelihood values in 0.0-0.3 range, while our model produces confidence in 0.0-1.0 range. This results in lower false positive and true positive rates for the former models than ours. $\mathcal{L}_{RAL}$ tends to produce overconfident predictions as noticeable from the high MAP(pa) scores corresponding to high $d_{PWCD}$ scores. On average, $\mathcal{L}_{ASL}$ loss is likely to be better performing than

Table 2: Comparison of performance of SDMs over a variety of multi-label learning losses on two tasks: species range prediction (and zero-shot range prediction) and geo-feature regression. The proximity-aware metrics proposed are defined in Section 4.2.

| Loss | Env. | Method | GBIF'21 | | GBIF'22 | | Zero Shot | | Geo Feature |
|------|------|--------|---------|---|---------|---|-----------|---|-------------|
| | | | MAP(pa)↑ | $d_{PWCD}$↓ | MAP(pa)↑ | $d_{PWCD}$↓ | MAP(pa)↑ | $d_{PWCD}$↓ | (Mean $R^2$) |
| $\mathcal{L}_{AN\text{-}full}$ | ✗ | SINR | 62.64 | 0.177 | 60.28 | 0.178 | - | - | 0.446 |
| $\mathcal{L}_{ME\text{-}full}$ | ✗ | SINR | 61.88 | 0.189 | 58.81 | 0.187 | - | - | 0.518 |
| $\mathcal{L}_{ASL}$ | ✗ | SINR | 67.11 | **0.152** | 64.55 | **0.155** | - | - | 0.619 |
| $\mathcal{L}_{RAL}$ | ✗ | SINR | 62.02 | 0.291 | 60.42 | 0.297 | - | - | 0.536 |
| $\mathcal{L}_{AN\text{-}full}$ | ✗ | SIREN(SH) | 61.99 | 0.203 | 60.67 | 0.215 | - | - | 0.562 |
| $\mathcal{L}_{ME\text{-}full}$ | ✗ | SIREN(SH) | 64.44 | 0.197 | 64.16 | 0.212 | - | - | 0.630 |
| $\mathcal{L}_{ASL}$ | ✗ | SIREN(SH) | 62.56 | 0.189 | 62.43 | 0.206 | - | - | 0.481 |
| $\mathcal{L}_{RAL}$ | ✗ | SIREN(SH) | 63.13 | 0.192 | 62.98 | 0.209 | - | - | 0.488 |
| $\mathcal{L}_{ASL}$ | ✗ | LD-SDM (ours) | **73.88** | 0.174 | **71.45** | 0.182 | 59.27 | 0.648 | 0.583 |
| $\mathcal{L}_{RAL}$ | ✗ | LD-SDM (ours) | **73.46** | 0.176 | **71.52** | 0.184 | 55.22 | 0.694 | 0.587 |
| $\mathcal{L}_{AN\text{-}full}$ | ✓ | SINR | 64.25 | 0.184 | 62.43 | 0.181 | - | - | - |
| $\mathcal{L}_{ME\text{-}full}$ | ✓ | SINR | 64.13 | 0.200 | 60.67 | 0.222 | - | - | - |
| $\mathcal{L}_{ASL}$ | ✓ | SINR | 67.23 | **0.172** | 69.49 | **0.179** | - | - | - |
| $\mathcal{L}_{RAL}$ | ✓ | SINR | 64.98 | 0.224 | 63.77 | 0.252 | - | - | - |
| $\mathcal{L}_{ASL}$ | ✓ | LD-SDM (ours) | **74.16** | 0.222 | **72.54** | 0.231 | 61.14 | 0.702 | - |
| $\mathcal{L}_{RAL}$ | ✓ | LD-SDM (ours) | **75.26** | 0.219 | **74.13** | 0.224 | 59.65 | 0.715 | - |

Table 3: Comparison of performance of our models using three different variants of LLaMA-2. The 70B variant achieves the best performance followed by the 7B and 13B variants.

| Text Encoder | encoding dim | GBIF'21 | | GBIF'22 | | Zero Shot | | Geo Feature | FLOPS |
|--------------|--------------|---------|---|---------|---|-----------|---|-------------|-------|
| | | MAP(pa)↑ | $d_{PWCD}$↓ | MAP(pa)↑ | $d_{PWCD}$↓ | MAP(pa)↑ | $d_{PWCD}$↓ | (Mean $R^2$) | TFLOPS |
| LLaMA-2-7B | 4096 | 71.24 | 0.177 | 70.25 | **0.178** | 56.27 | 0.719 | 0.579 | 0.174 |
| LLaMA-2-13B | 5120 | 70.11 | 0.184 | 69.09 | 0.193 | 55.70 | 0.732 | 0.563 | 0.342 |
| LLaMA-2-70B | 8192 | **73.88** | **0.174** | **71.45** | 0.182 | **59.27** | **0.648** | **0.583** | 1.848 |

Table 4: Comparison of taxonomic mapping performance of our models using the three different variants of LLaMA-2 for taxonomic encoding. These performances are reported on the combined presence-only dataset from GBIF for the years 2012-2022. It is worth noting that LLaMA-2-13b has a lower performance than LLaMA-2-7b, possibly due to differences in their training data.

| Taxonomic Level | LLaMA-2-7B | | LLaMA-2-13B | | LLaMA-2-70B | |
|-----------------|------------|---|-------------|---|-------------|---|
| | MAP(pa)↑ | $d_{PWCD}$↓ | MAP(pa)↑ | $d_{PWCD}$↓ | MAP(pa)↑ | $d_{PWCD}$↓ |
| Class | 97.10 | 0.117 | 95.2 | 0.204 | 97.52 | 0.163 |
| Order | 56.15 | 0.450 | 55.94 | 0.472 | 58.31 | 0.414 |
| Family | 38.18 | 0.659 | 36.47 | 0.688 | 40.13 | 0.632 |
| Genus | 24.95 | 0.710 | 24.97 | 0.711 | 27.86 | 0.692 |

the other losses over all the models. For SINR, $\mathcal{L}_{ASL}$ performs the best as compared to the other losses by a large margin. This is also evident in the geo-feature regression task. Our model has also produced competitive scores on the task of geo-feature regression. The coordinate-only models perform nearly as well as models incorporating environmental features. So, dropping environmental features could result in faster training.

**Qualitative Examples.** In Figure 4, we show global-scale species distribution maps generated for five bird species. To make fair comparisons, we used the best-performing coordinate-only models to ensure no bias is added to models from the environmental features. Although SINR and SIREN(SH) could generate species distribution maps at any spatial resolution, they tend to learn a very low-frequency function over the sphere as evident from the blob-like maps. SIREN(SH)

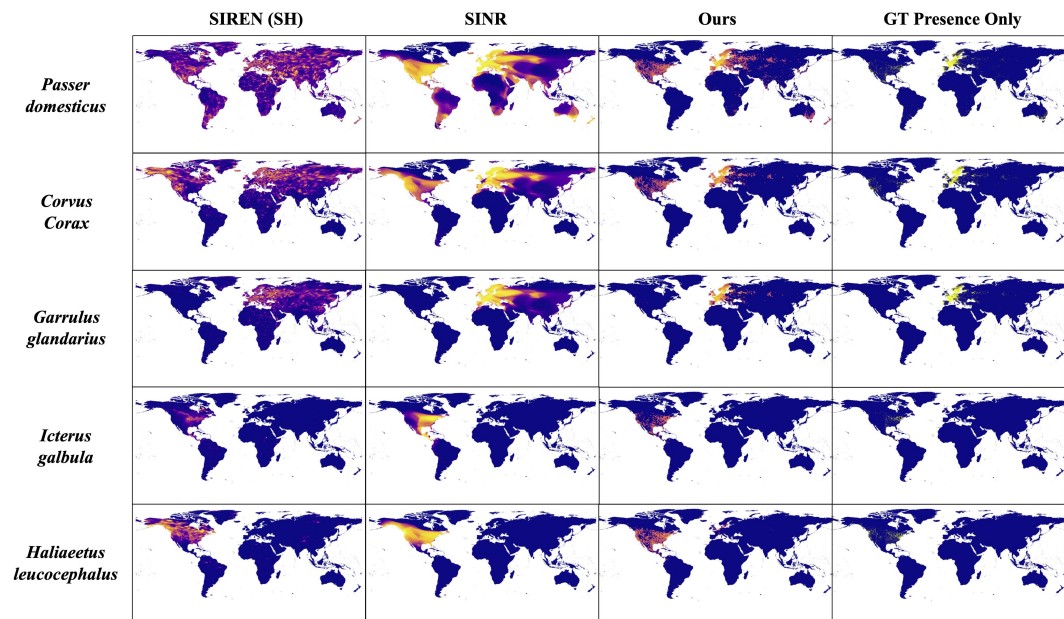

Figure 4: **Predicted Range Maps (Probability Values)**. The predictions are generated at a spatial resolution of 0.2 degrees. For the SIREN(SH) and SINR models, predictions are generated at the center of each raster cell. The visualizations show that our model performs better than other models at resolving higher frequencies over the sphere and localizing the species distribution.

produces a lot of false positive predictions possibly due to the limitation of the number of Legendre polynomials it can fit. Our model on the other hand learns to predict species ranges at a more fine-grained level. This is evident from the visualization of the geographic location features learned by the location encoder of each model (shown in Appendix D). Fine-grained geographic features are important for better geolocalization of species. SINR and SIREN(SH) tend to learn similar features over large geographic regions while LD-SDM produces higher variability in the features at a finer scale. The learned features could be used for other ecological mapping tasks that require geography-aware features.

**Effect of LLaMA-2 variants on LD-SDM.** We assess the impact of using different variants of LLaMA-2 for LD-SDM in Table 3. In particular, we compare the 7B, 13B, and 70B variants. Surprisingly, the 13B variant performs the worst. This is possibly due to the differences in the training data of the variants. Empirically, this can be explained by examining the T-SNE plots of text embeddings. In Appendix A, we present T-SNE plots along with mean intra-cluster and inter-cluster distances of text embeddings. Ideally, both intra-cluster and inter-cluster distances should be high to ensure separability within a particular taxonomic class and between taxonomic classes respectively. 13B variant has the poorest separability at every taxonomic rank. The 7B and 70B variants have almost identical separatabilities. The impressive performance of the 70B variant can be attributed to its high dimensionality of the species-specific encoding. Although 70B has noticeably higher computational overheads, it is compensated by precomputing the text embeddings before training. Overall, all the variants can perform better than the state-of-the-art.

**Taxonomic range mapping using LLaMA-2 variants.** We perform range prediction over the taxonomic ranks: *class*, *order*, *family*, and *genus*. The procedure is the same as for generating species range maps. We precompute the embeddings for each rank as shown in Figure 1. The range map for the rank *class* represents the unconditional distribution of finding any bird *species* in the world. As one moves down in the hierarchy, the range maps become more and more fine-grained. As expected, the performance of LD-SDM drops as one moves down in the hierarchy (Table 4). This can be explained by examining the increasing $d_{\text{PWCD}}$ scores. The T-SNE plots of the embeddings of various ranks in Appendix A help in better understanding the increase in false

positive rates as one moves down in the hierarchy. As it can be seen, the intra-cluster distances decrease substantially when moving down in the hierarchy. As a result, it becomes challenging for the model to distinguish between birds for example of *genera* belonging to the same *family*.

**Spherical Harmonics are important for SDM** We compare the performance of LD-SDM when using different backbones in the location encoder network (Table 5). All the models are trained from scratch using the LLaMA-2-70B variant. We use the same training inputs including the environmental features. This is done since positional encodings are implicitly incorporated by the Segformer and DINOv2 architectures. All the models are trained using the $\mathcal{L}_{\text{ASL}}$ loss. Notice that SFNO gives the best performance over Segformer and DINOv2. With SFNO, we observe a significant gain in performance and a reduction in false positives. Overall, a robust geographic feature extractor is beneficial for SDM and SFNO proves to be one such architecture.

Table 5: Comparison of performance of three different backbones used in the parameterization network. All the models are trained with the same loss with environmental features included in the inputs.

| | **Params** | **GBIF'21** | | **GBIF'22** | |
|---|---|---|---|---|---|
| Backbone | M | MAP(pa) $\uparrow$ | $d_{PWCD} \downarrow$ | MAP(pa) $\uparrow$ | $d_{PWCD} \downarrow$ |
| Segformer (Xie et al., 2021) | 20.81 | 71.64 | 0.247 | 70.54 | 0.255 |
| DINOv2 (Oquab et al., 2023) | 41.34 | 72.19 | 0.232 | 72.01 | 0.240 |
| SFNO (Bonev et al., 2023) | 19.19 | **74.16** | **0.222** | **72.54** | **0.231** |

## 5 LIMITATIONS

As pointed out in (Cole et al., 2023), the crowdsourced data is biased towards regions with heavy human traffic. Most often, rich biodiverse regions are underrepresented limiting the performance of SDMs. In this work, we do not explicitly deal with the observation bias in presence-only SDM. However, our goal was to incorporate taxonomic information into SDM to enhance unseen species discovery.

Previous studies have evaluated species SDMs using expert range maps, which are only available for a limited number of species. Our proposed evaluation metric allows for the evaluation of SDMs using crowdsourced data, which provides a more comprehensive representation of numerous species. However, we note that none of these methods are error-free. Yet, with our proposed evaluation metric, innovative methods could be developed for the semi-automatic evaluation of SDMs in the future, to strike a balance between accuracy and efficiency.

## 6 CONCLUSIONS

We introduced a novel approach for species distribution modeling that uses a large-language model to generate a representation of a species. This provides flexibility to generate distributions at different levels of the taxonomic hierarchy and for unseen species. In addition, we introduced a new evaluation metric, Probability Weighted Chamfer Distance. This metric computes the false positive rate in species range mapping tasks in a proximity-aware manner. While our focus was on species distribution modeling, the integration of large-language models points to some intriguing potential future use cases. It might be possible, for example, to generate distribution models from arbitrary text, such as "a large white bird" or "a flock of red-winged blackbirds". This is potentially more useful for non-expert users. Finally, alternative text prompts could be explored as future work which may better reason about a species and its dependence on a particular geographic location.

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

## A    SPECIES-SPECIFIC EMBEDDINGS

As described in the paper, we encode the taxonomic information of species in the form of text. Text is a dictionary consisting of key-value pairs representing each taxonomic rank. We use `hugging-face`'s library `transformers` to obtain all the LLaMA-2 variants used in the paper. We separately compute and save the embeddings of the key-value pairs for each variant. In Table 6, we report the number of tokens generated by the LLaMA-2 variants for each taxonomic rank.

Table 6: Total tokens generated by LLaMA-2 for each taxonomic rank. The dimensionality of each of these tokens depends on the type of LLaMA-2 variant used.

| Taxonomic Rank | #tokens | #classes | #total tokens |
|---|---|---|---|
| order | 13 | 40 | 520 |
| family | 19 | 211 | 4009 |
| genus | 25 | 1393 | 34,825 |
| species | 34 | 4141 | 140,794 |

We create T-SNE plots in 2-D for visualizing the embeddings at each rank resulting from the variants (Figure 5). To create the plot, we do an average pool across the number of tokens' dimension for each text embedding. We then use `scikit-learn` library to compute the T-SNE embeddings for each species and at every taxonomic rank. We color code the embeddings based on the taxonomic rank one level higher than the current rank. For example, the T-SNE embeddings for the rank *family* are color-coded by their *order*. Doing this provides a simple intuition for interpreting the separatability of species and taxonomic ranks.

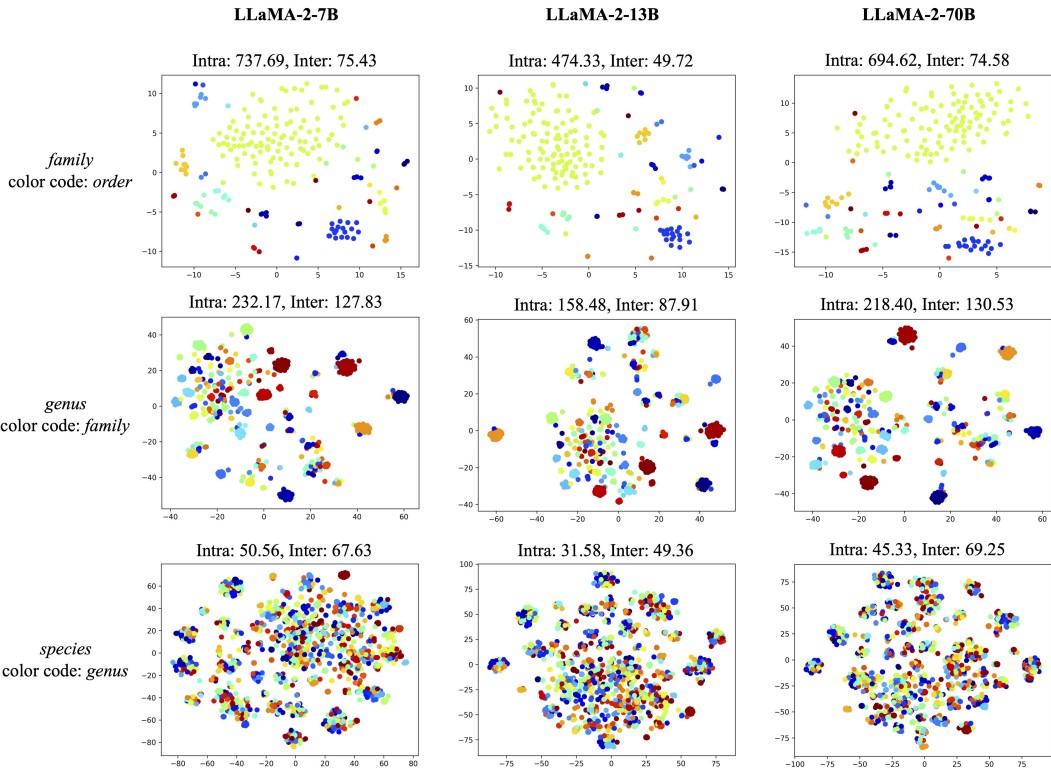

Figure 5: **T-SNE Plots of Text Embeddings**. For each variant and taxonomic rank, we compute the T-SNE embeddings of all the text tokens generated. We do average pooling over the token dimension before computing T-SNE. These plots highlight the separatability degree of each variant over each taxonomic rank.

We compute average inter-cluster and intra-cluster distances based on the color coding for each taxonomic rank. This is also done using the `scikit-learn` package. The inter-cluster distance represents the average dissimilarity between *species* (or any other rank) of different classes. The intra-cluster distance represents the average dissimilarity between *species* (or any other rank) of the same class. Intuitively, both distances should be high enough to ensure good separatability for SDM. As we see, the distances decrease as we move down in the hierarchy. This is expected since the distribution of each species becomes more and more fine-grained. The 13B variant has abruptly low distances compared to the other variants suggesting poor performance in SDM. This can possibly be due to differences in training data and the training procedure between the variants.

## B  PROBABILITY WEIGHTED CHAMFER DISTANCE

We proposed the Probability Weighted Chamfer Distance (PWCD) which is a spatially-aware proxy for false-positive rates when comparing two probability maps. This distance could also be used in other metrics such as precision, MAP, or AUC to make them spatially aware. In Listing 1, we describe NumPy-style pseudocode for computing this distance metric.

We use the `xrspatial`'s `proximity` function for computing the nearest-neighbor distance map given the binary ground-truth range map. All the distances are measured in pixel units using Euclidean distance. The minimum possible distance is 0 which is for the positive pixels having *species* presence. The maximum possible distance depends on the spatial resolution of the range map. In our case, the spatial resolution of $0.2^o$ gives a maximum distance of 2012.46. It is noteworthy that this distance function could be replaced with any other distance function for example the haversine distance.

Listing 1: NumPy-style pseudocode for computing $d_{\text{PWCD}}$

```
import xarray as xr
from xrspatial import proximity
import numpy as np

#y_true[h, w]: true binary range map
#y_pred[h, w]: predicted range map
#a: temperature parameter

def PWCD(y_true, y_pred, a):
    xr_true = xr.DataArray(y_true)
    prox_agg = np.array(proximity(xr_true))
    pwcd = 1 - np.exp(-a*y_pred*prox_agg)
    fps = np.sum(pwcd)
    neg = np.sum(y_true == 0)
    return fps / neg
```

We visualized the pixel-level $d_{\text{PWCD}}$ map in Figure 6. Notice that the predictions spatially close to the true presence of *species* are weighted less than the predictions that are distant. This weighting is due to the nearest-neighbor distances of the pixels making the metric spatially-aware. The chamfer distance map describes the weighting of predictions as a function of distance from the nearest presence pixel. The weight gradually increases as the distance from the nearest presence pixel increases. Note that the nearest-neighbor distance map differs from the chamfer distance map, which is in the range of [0,1]. The chamfer distance map is computed using the nearest-neighbor distance map, using the formula for $d_{\text{PWCD}}$ and a constant likelihood term equal to one.

In Figure 7, we show the effect of temperature parameter $\alpha$ on the $d_{\text{PWCD}}$ metric. The figure shows chamfer distance as a function of nearest-neighbor distance at a fixed likelihood probability of one and varying values of $\alpha$. The parameter $\alpha$ controls the exponential weighting of the nearest-neighbor distance in the metric. Increasing $\alpha$ gradually increases the weight of nearby pixels. The metric starts penalizing more of the nearby pixels in the computation of false positive rates. A large value of $\alpha$ will make the metric more sensitive to spatial shifts in the predictions while a lower value of $\alpha$ will be more robust towards spatial shifts. However, an extremely low value of $\alpha$ will make the metric ignore distant false positives. As a result, an optimal value of $\alpha$ is essential to evaluate SDMs.

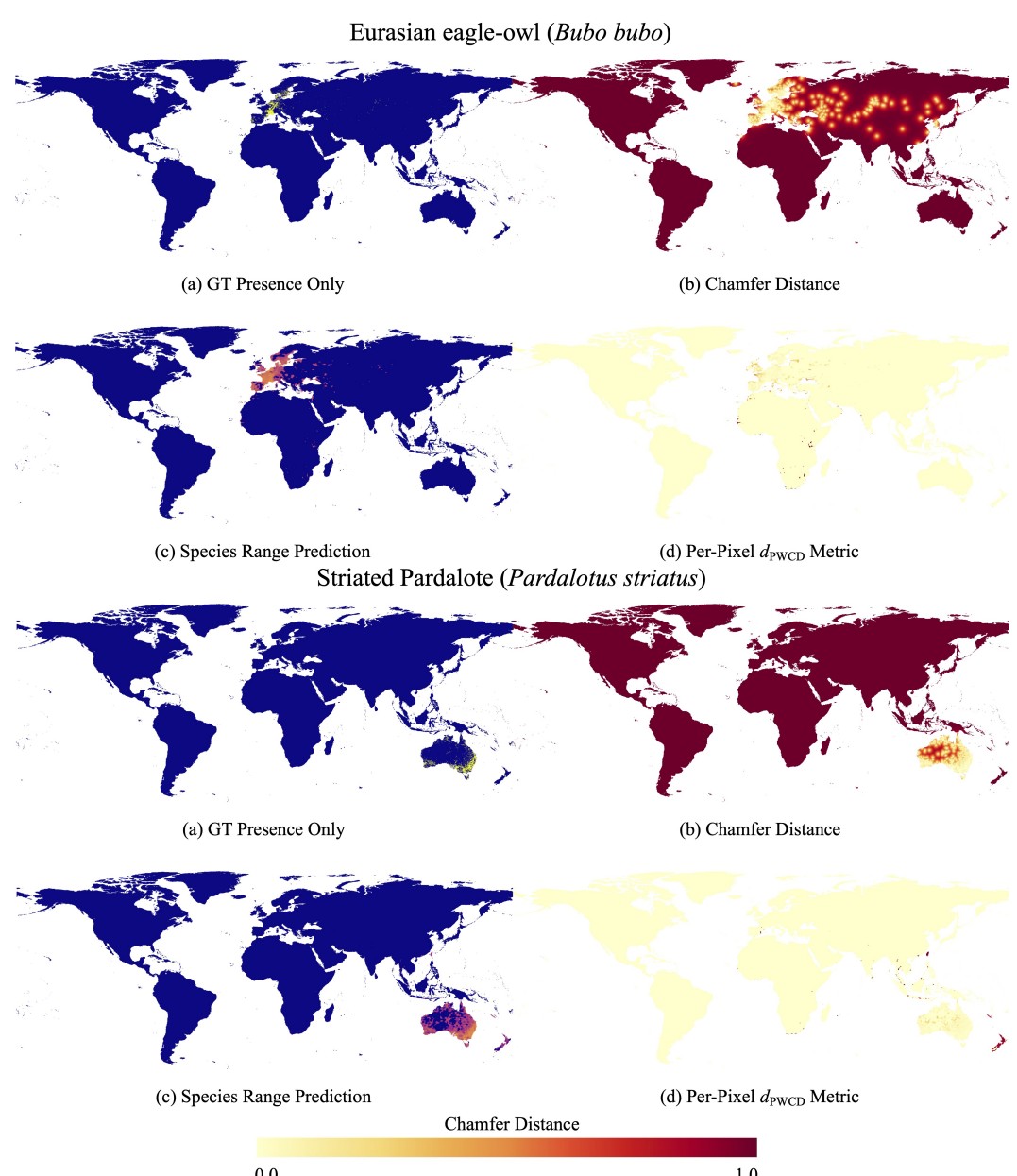

Figure 6: **Probability Weighted Chamfer Distance**. Using the presence-only ground-truth maps, the nearest-neighbor distance map is computed. Finally, $d_{\text{PWCD}}$ is calculated using the per-pixel activation values from the range map predicted by an SDM and the nearest-neighbor distance map. The distance metric for computing the nearest-neighbor distance map is the Euclidean Distance in pixel units.

In order to evaluate the localization of species distributions, we have plotted $d_{\text{PWCD}}$ scores with varying $\alpha$ in Figure 8. The first plot shows the relationship between $d_{\text{PWCD}}$ and the nearest neighbor distance for varying $\alpha$ values. By analyzing this graph, one can determine the optimal value of $\alpha$ for their specific application. For our purposes, we selected $\alpha = 0.1$, as it is robust to spatial shifts and accurately detects the distribution of species.

In the second plot, we have shown the $d_{\text{PWCD}}$ scores as a function of the likelihood probability, at a constant nearest neighbor distance of 100 pixels. As the value of $\alpha$ increases, the metric starts

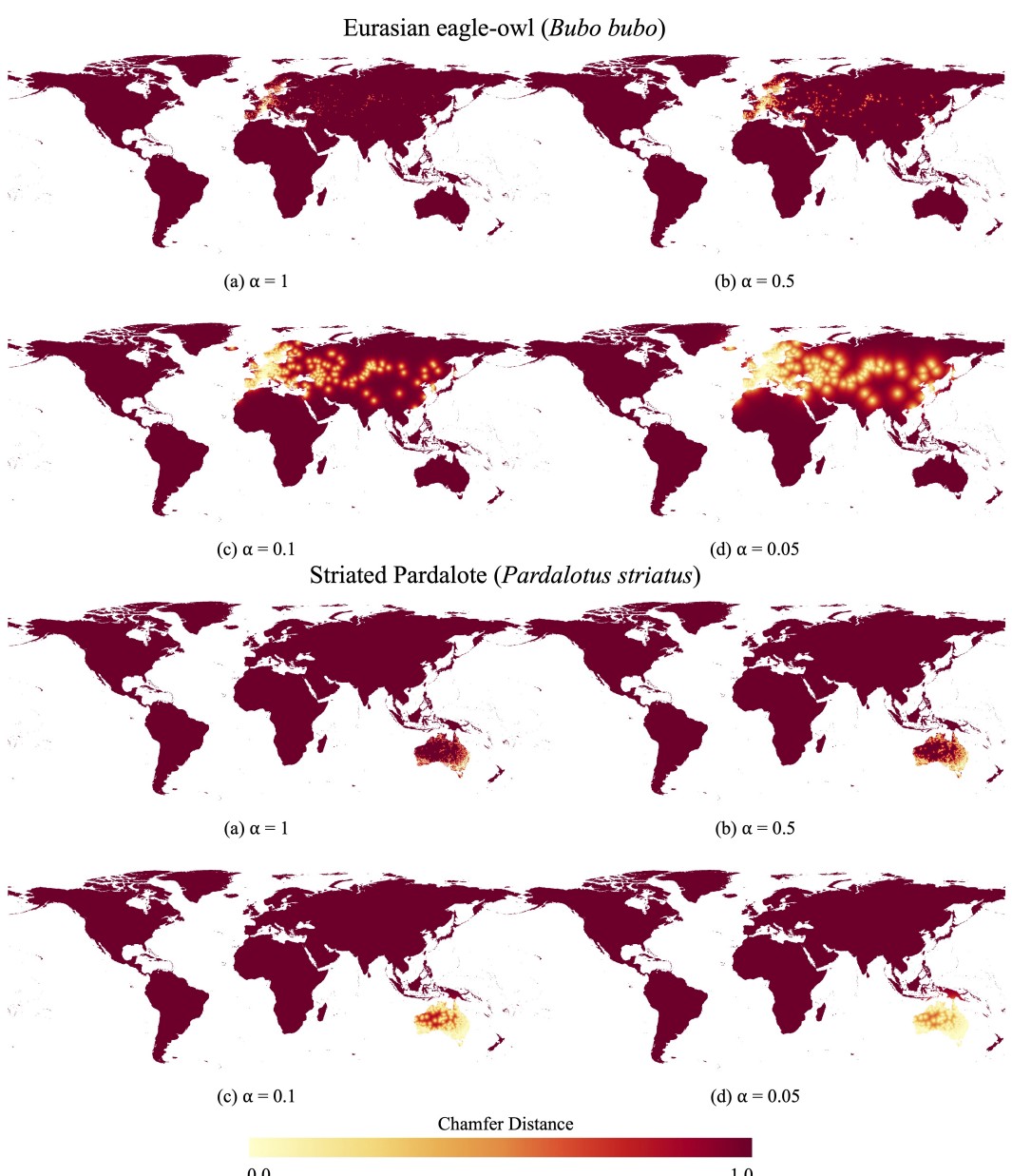

Figure 7: **Effect of $\alpha$ on Chamfer Distance**. With an increasing value of $\alpha$, metric $d_{\text{PWCD}}$ becomes more sensitive to prediction shifts. Lower values of $\alpha$ make the metric $d_{\text{PWCD}}$ more robust to shifts at the cost of losing assessing spatial localizability in an effective sense.

penalizing low likelihood values. This gives us more control over the trade-off between the distance and likelihood scores, allowing us to determine the optimal $\alpha$ value for our application.

## C  TRAINING

Below we describe the details of the datasets used. We also describe the implementation details of the models and the losses. Each training run was done on a single NVIDIA A40 GPU.

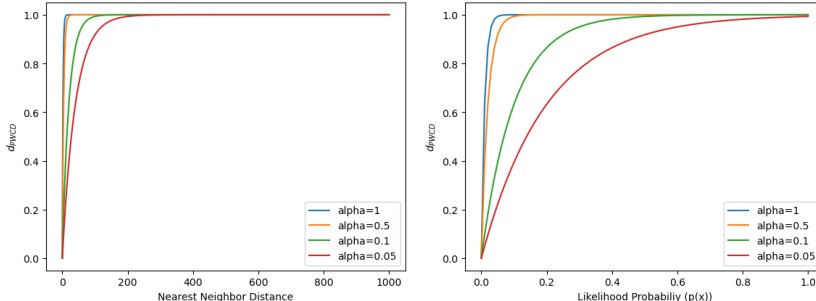

Figure 8: **Sensitivity of** $d_{\mathbf{PWCD}}$. We show the variation of $d_{\mathrm{PWCD}}$ as distance to ground truth observation increases (left). We also show the variation of $d_{\mathrm{PWCD}}$ with likelihood probability at a fixed distance of 100 pixels from ground-truth (right).

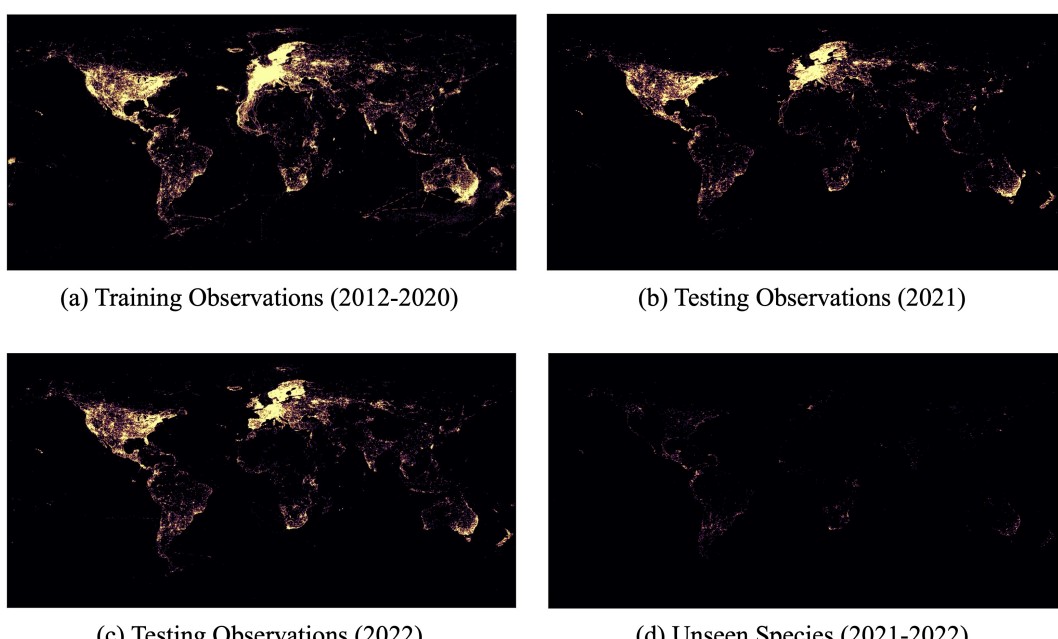

(a) Training Observations (2012-2020)    (b) Testing Observations (2021)

(c) Testing Observations (2022)    (d) Unseen Species (2021-2022)

Figure 9: **Histogram of Locations of Observations**. We show the visualization of observations of the datasets considered in this study at a spatial resolution of $0.2^o$.

## C.1   DATA

Our training and evaluation data was collected using various citizen science platforms such as the iNaturalist (inaturalist.org) (Van Horn et al., 2018) and eBird (ebird.org) (Sullivan et al., 2009). The maps in Figure 9 display the geographical locations of the observations for each dataset across the globe. They were compiled using the GBIF (gbif.org) (Robertson et al., 2014) platform, which hosts occurrence records of species from several citizen science platforms. For compilation, we used the following conditions:

1. All observations must have valid labels for each taxonomic rank and geographical location values.
2. All observations must have the *research grade* status label.

To avoid bias, we excluded species with less than 100 observations. The evaluation datasets only contain observations from iNaturalist and eBird. For zero-shot evaluation, we selected the species that were not present in the training dataset. Further, we filtered out species with more than 1000 observations to focus on rare species.

We use bioclimatic variables and elevation data as environmental covariates in the SDM. The variables are derived from the WorldClim dataset (worldclim.org) (Fick & Hijmans, 2017), which are at a spatial resolution of $0.01^o$. The dataset is resampled to $0.2^o$ resolution using bilinear interpolation. In total, the dataset contains 20 different variables (channels). We use nine different geographic features in the geo-feature regression task as done by (Cole et al., 2023). These include above-ground carbon, elevation, etc. All the features are resampled to $0.2^o$ resolution.

Table 7: Training settings for SINR

| Config | Value |
| --- | --- |
| backbone | ResNet-style MLP |
| encoding dim | 256 |
| depth | 4 |
| input encoding | sin-cos |
| optimizer | AdamW |
| base learning rate | 5e-4 |
| learning rate scheduler | exponential decay |
| batch size | 2048 |
| epochs | 10 |

Table 8: Training settings for SIREN(SH)

| Config | Value |
| --- | --- |
| backbone | SIREN |
| encoding dim | 256 |
| depth | 4 |
| input encoding | spherical-harmonics |
| L | 20 |
| optimizer | AdamW |
| base learning rate | 1e-4 |
| learning rate scheduler | exponential decay |
| batch size | 2048 |
| epochs | 20 |

Table 9: Training settings for LD-SDM

| Config | Value |
| --- | --- |
| backbone | SFNO |
| encoding dim | 128 |
| SFNO blocks | 2 |
| multi-label classifier | hidden_dim: 64 |
| | num_heads: 16 |
| optimizer | AdamW |
| base learning rate | 1e-5 |
| batch size | 1 |
| accumulate grad batches | 64 |
| epochs | 10 |

## C.2 MODELS

We experimented with three different SDMs: 1) SINR (Cole et al., 2023); 2) SIREN(SH) (Rußwurm et al., 2023); LD-SDM (ours). For the baseline models, we used the same hyperparameters as reported by authors. Tables 7, 8 and 9, show the hyperparameters used for each model. We use $L = 20$ for SIREN(SH) since experiments with larger $L$ were computationally infeasible.

| (a) SIREN (SH) | (b) SINR | (c) LD-SDM (ours) |

Figure 10: **Features Learned by Location Encoder**. We show the visualization of location embeddings learned by the location encoder of various models. For visualization, the embeddings are projected to a 3-dimensional space using Independent Component Analysis. SINR learns a low-frequency representation of location while SIREN(SH) and LD-SDM learn to localize at a fine-grained level.

### C.3 LOSSES

We evaluated four different multi-label learning losses in our experiments, namely: 1) AN-full (Cole et al., 2023); 2) ME-full (Zhou et al., 2022); 3) ASL (Ridnik et al., 2021); 4) RAL (Huang et al., 2023). For AN-full and ME-full, we used a positive weight of 2048. The pseudo-negative sampling is done according to (Cole et al., 2023), by using a spherical buffer around each observation. In the case of ASL and RAL, we used 0 and 4 for $\gamma^+$ and $\gamma^-$ respectively. In RAL, we set $\lambda$ to 1.5.

## D LOCATION EMBEDDINGS

We present the location-specific features learned by the SDMs in Figure 10. The location embeddings displayed are the output of the location encoder network for each SDM. We use Independent Component Analysis (ICA) to project the high-dimensional embeddings to a 3-dimensional space. All the maps are generated at a spatial resolution of $0.2^o$. SINR produces a smooth map indicating that it has failed to learn high-frequency functions. SIREN(SH) produces noisy location features spread across the globe. LD-SDM produces a map that resembles the distribution of training data. We see that in the locations where observations are absent, LD-SDM has learned similar features.

## E LIGHTWEIGHT SFNO PERFORMS SURPRISINGLY WELL

In Table 10, we show the effect on the performance of species range prediction due to different components of the SFNO. It is noteworthy that an encoder dimension of 128 for SFNO yields results on par with an encoder dimension of 256 while reducing computational complexity. On the other hand, having fewer encoder layers in SFNO is beneficial since performance gain from adding additional encoder layers is marginal. In conclusion, a lightweight SFNO is a strong geographic feature extractor that can be used in various downstream global-scale geospatial tasks.

Table 10: SFNO ablation in LD-SDM. We experiment on various values of hidden encoder dimension and the number of encoder layers in SFNO. All experiments run with the $\mathcal{L}_{\text{ASL}}$ loss and the LLaMA-2-70B variant.

| | GBIF'21 | | GBIF'22 | |
| --- | --- | --- | --- | --- |
| Encoder Dim. | MAP(pa) $\uparrow$ | $d_{PWCD} \downarrow$ | MAP(pa) $\uparrow$ | $d_{PWCD} \downarrow$ |
| 64 | 68.55 | 0.147 | 67.28 | 0.149 |
| 128 | 73.88 | 0.174 | 71.45 | 0.182 |
| 256 | 73.67 | 0.178 | 71.82 | 0.188 |
| Encoder Layers | MAP(pa) $\uparrow$ | $d_{PWCD} \downarrow$ | MAP(pa) $\uparrow$ | $d_{PWCD} \downarrow$ |
| 2 | 73.88 | 0.174 | 71.45 | 0.182 |
| 3 | 73.39 | 0.171 | 72.11 | 0.174 |
| 4 | 73.67 | 0.174 | 72.00 | 0.180 |

