# OpenReview forum: "LD-SDM: Language-Driven Hierarchical Species Distribution Modeling"
_ICLR.cc/2025/Conference — Submitted to ICLR 2025_

### Official Review · Reviewer_nCqY · 2024-10-19

**Soundness:** 2
**Presentation:** 3
**Contribution:** 3
**Rating:** 5
**Confidence:** 4

**Summary:**

This work addresses the problem of species distribution modelling, i.e., estimating the spatial distribution of a species from a limited set of point observations. A new model, LD-SDM, is proposed that directly predicts a probability for a discretized set of locations indicating if a species is present there or not. The model is conditioned on a set of environmental features and uses a pre-trained language model to encode the identity of the species of interest. Results are presented on a new dataset adapted from public crowdsourced data where the proposed model outperforms existing work.

**Strengths:**

The main strengths are:
[S1] This is the first SDM approach, that this reviewer is aware of, that incorporates language. However, only a restrictive subset of all possible input text is used, i.e., a string encoding the taxonomic hierarchy for the species of interest
[S2] Unlike existing methods that typically predict for a single location of interest, the proposed model makes a prediction for all locations simultaneously for one species at a time
[S3] In addition to the standard losses used in this problem space, the authors also evaluate different training losses from the single positive multi label literature
[S4] The quantitative results in Table 2 demonstrate a non-trivial improvement over existing state of the art methods

**Weaknesses:**

The main weaknesses are:
[W1] The central assumption that “species belonging to the same genera tend to be found in similar locations” is a big assumption and not backed up with evidence. See Q1 below for a request for more information about this.
[W2] It is not clear if it is necessary to use an LLM to encode the taxonomy/hierarchy. A simpler approach could be to train a per-species embedding as in SINR. For unseen species at inference time, one could just use the average embedding of all species from the train set that belong to the same genus. At the very least, an ablation akin to this would allow the readers to understand if the LLM adds value.
[W3] A new training and evaluation dataset is proposed. However, it is not clear why the data from Cole et al. ICML 2023 is not used. Line 72 states that hand crafted range maps are “not widely available”. However, this is exactly the type of evaluation data used in Cole et al.
[W4] The proposed evaluation metric would appear to be very sensitive to missing presences in the evaluation data. As indicated in the paper, species observation data is highly spatially biassed. Thus, only evaluating models on locations where data has been observed runs the risk of incorrectly penalising correct predictions from a model when there may in fact actually be presences for the species in that location, but they are just not in the evaluation dataset. The advantage of the evaluation protocol used in Cole et al. is that the evaluation data is spatially dense.

Addition minor comments, that do not require a response in the rebuttal:
L25 “remote-sensing task” -> SDM is not a remote sensing task
L27 goal is to “produce large-scale range maps” -> the SDM problem represents a wider set of tasks than just range estimation, e.g., abundance
L51 It is not clear what this sentence means, i.e., there are models that have been trained with location only, location and environmental features, and location and remote sensing images. It should be rephrased
L71 “presence-only” data is introduced here for the first time and not defined
L142 “n” is used for the dimensionality of the environmental features and “N” is used for the number of training observations
L344 should make it clear in the text that most of the data in GBIF is actually from iNaturalist or eBird
L345  Van Horn et al. 2018 is the incorrect reference, i.e., it is the iNaturalist 2017 image classification dataset, not the iNaturalist website
L361 this spatial binning will not result in an even distribution around the globe, it would be better to use H3 cells so the poles are not overrepresented. Perhaps that is what is being done, but it is not clear?
L516 should note in the limitations the big assumption that taxonomic is a good proxy for range similarities
Fig 9 the spatial distribution of unseen species is very different from the others. This should be commented on

**Questions:**

[Q1] Line 35 states that “species belonging to the same genera tend to be found in similar locations”. This seems like a big assumption, as I would have assumed that the most common reason historically that new species were formed is because populations became geographically isolated from one another. It makes sense that they would have an affinity for the same habitats, but it is not clear if they would be found in the same location.  Is there evidence to support this claim e.g., from the literature or your training data?

[Q2] How is the zero-shot evaluation performed? Is the full taxonomic name for the unseen species just provided to the model at inference time?

[Q3] Can you provide statistics of how many of the species in the test sets are from the same genus as those in the train set. Also, report on average how many species are in each genus in the train set.

---

### Official Review · Reviewer_zn3q · 2024-11-01

**Soundness:** 2
**Presentation:** 2
**Contribution:** 1
**Rating:** 3
**Confidence:** 3

**Summary:**

The authors tackle the problem of predicting the geographic range of a biological species given a set of opportunistic observations. They do so by leveraging a model the uses spherical harmonics to better fit functions defined on the Earth’s surface. Their main novelty is to encode the species with their full taxonomic name, thus allowing species that share part of their life tree to be encoded closer together.

**Strengths:**

1) The paper is written and structured.
2) The usage of an adapted version of the chamfer distance is interesting, since eveluation metrics are typically a bottleneck in developing SDMs.
3) The results show that their method is competitive against two competing methods.

**Weaknesses:**

1) I was a bit surprised by their choice of main contribution: the use of full taxonomic names based on language models. This seems to assume that taxonomically similar species tend to have similar ranges, but I could not find a justification for this. I’m not an ecologist, but this seems to go counter the idea that speciation is promoted by the geographic isolation of populations.
2) Although I do understand that such an approach may be useful for obtaining range maps of higher-level taxa based on training only with species-level data, I could not grasp the rationale of why this would help performing zero-shot with unseen species. Why would the species name itself carry any information about its range?
3) The experimental setup does not manage to show the value of the different elements in the method: is there an advantage to using the full taxonomic name vs the binary name? Does the binary name provide better results that a one-hot encoding of the species? Even the last conclusion, “Spherical Harmonics are important for SDM”, is not really based on the provided results, since it is not clear why the models compared in Table 5 are actually comparable, or that they differ only in the use of spherical harmonics.

**Questions:**

As follows from the weaknesses:
1) What is the justification for assuming that taxonomically similar species tend to have similar geographic ranges, given that speciation is often promoted by geographic isolation?
2) How does using full taxonomic names facilitate zero-shot predictions for unseen species, especially considering that species names may not inherently convey information about geographic range?
3) Are there benefits of using the full taxonomic name over the binary name, or of using the binary name compared to a one-hot encoding? Additionally, how does the conclusion that "Spherical Harmonics are important for SDM" follow from the results in Table 5?

---

### Official Review · Reviewer_GZqH · 2024-11-03

**Soundness:** 2
**Presentation:** 4
**Contribution:** 4
**Rating:** 3
**Confidence:** 4

**Summary:**

This paper considers the important problem of large-scale species distribution modeling. Different from prior works, this paper uses an LLM to encode taxonomic information for each species. The proposed model is also capable of zero-shot prediction, which is a novel capability. This work also introduces a novel evaluation metric that is meant to be more sensitive to spatial proximity when considering prediction errors. The paper also curates new training and evaluation datasets. Generally, the paper is interesting and valuable, but there are some experimental methodology issues that should be resolved.

**Strengths:**

* The paper is generally clearly written and the figures are well-designed.
* The incorporation of language into species distribution modeling is novel to the best of my knowledge.
* The proposed method significantly outperforms other methods (at least on the new metric introduced in this paper - see comments on this below).
* Comprehensive evaluation of different combinations of prior methods and losses (including the $ASL$ and $RAL$ multi-label losses, which had not been previously studied for this problem).
* Separating out evaluation on "unseen" and "rare" species is and interesting and valuable idea.
* The paper introduces a new proximity-aware metric which seems interesting and intuitively reasonable.
* The hyperparameters are clearly specified and there is enough information to reproduce the results of the paper.
* Species distribution modeling is an interesting and important problem.

**Weaknesses:**

Missing pieces:
* Computational resources seem to be a major confounder (in terms of model sizes and potentially training times). Can you explicitly compare the sizes  and # training steps for different models to contextualize the performance differences we observe between the proposed model and prior works?
* We know prompt engineering is very important for language models. Were there any attempts at prompt engineering? What if we just use the species name without any other taxonomic information? There doesn't seem to be any evidence that the proposed text encoding is a better choice than other simple alternatives.
* A new metric was proposed, but the problem it is meant to solve was not demonstrated to exist. (Though I agree that intuitively it seems reasonable that the problem might exist.) Finding a few nice examples would fix this. It was also not demonstrated that the new metric is providing different information than existing ones in practice. For instance, does the new metric rank models differently than previous commonly used metrics? (Showing pre-existing metrics e.g. mAP alongside the new metrics would help.)
* This work only makes one direct comparison to other methods using existing metrics: the Geo Feature results in Table 2. In that case, existing models outperform the proposed model. This raises the concern that the proposed model might underperform prior models on other pre-existing tasks as well. Since the newly proposed metric is not yet accepted as the gold standard, it is important to demonstrate how the proposed model performs on prior metrics as well.
* How does the proposed metric handle the case of "vagrants" - species that are spotted far from their normal ranges? It seems like this is a problematic case (especially for birds!) that is not discussed.
(Apologies if any of these are not missing and I simply missed it, please point it out if so!)

Slightly misleading or unclear bits:
* L42-50 seems to indicate that the framework of Mac Aodha et al. cannot use "species-specific rich information". Is this true? What would stop their model from using fixed species embeddings (e.g. those based on taxonomic string embeddings) and learning the location encoder? Or using some sort of regularization to encourage learned species embeddings to have relative distances similar to those based on taxonomic string embeddings? It's not clear to me that the "reformulation" of the species distribution problem is necessary to achieve these goals, as is suggested by the paper.
* L184: I'm not quite sure I'd characterize the approach in this paper as "natural language" - the paper focuses on a highly constrained and artificial sort of language (taxonomic strings).
* "To avoid bias, we excluded species with less than 100 observations." What kind of bias are we talking about, and how does this avoid it?

Minor comments (no need to respond):
* The citation for iNaturalist (the platform) should probably be www.inaturalist.org instead of Van Horn et al. (which is a dataset derived from iNaturalist data) .
* L102: Strange capitalization for MacKenzie et al.
* L105: "With no expert knowledge" - I'd clarify that no expert knowledge is required to capture the raw data, but it is required to identify the species.
* Why not include common names? Might be closer to the distribution the language model was trained on.

**Questions:**

Since my questions closely parallel weaknesses, they have been stated in the "Weaknesses" section.

**Details Of Ethics Concerns:**

This is not a major point, but the paper should somewhere advise caution on potential misuses of the model, both by bad actors and by good actors who might trust the model results when they should not.

---

### Official Review · Reviewer_3Pji · 2024-11-04

**Soundness:** 2
**Presentation:** 2
**Contribution:** 2
**Rating:** 5
**Confidence:** 3

**Summary:**

The authors introduce a reformulation of species-distribution modeling to allow species embeddings via LLMs and introduce a new metric that takes spatial proximity into account.

**Strengths:**

**(S1)**: The problem tackled by this paper is underexplored in most ML works. The proposed solution appears to be straightforward to implement.

**(S2)**: The method can handle zero-shot prediction for species not seen in training data. This is a valuable advantage as a new classifier doesn’t need to be re-trained for each new query species.

**(S3)**: The new metric introduced accounts for spatial proximity which makes a more fine-grained analysis of performance possible.

**Weaknesses:**

**(W1)**: No theoretical guarantees are given for the proposed metric. Since the authors are introducing a new metric that is posited to be more robust to variations in location, it would be helpful to establish the bounds of tolerance for variability as well as formal guarantees. Nothing as presented about the metric makes it specific to species-distribution modeling (SDM), and so the authors must be mathematically specific about where and how this metric should be used.

**(W2)**: Lack of clarity in problem formulation. Why is the region variable $r$ discretized (section 3.1)? It would be more useful if $r$ was allowed to continuously vary which would more realistically model geospatial coordinates (i.e. latitude, longitude). One can still model the problem as $p(r|y,l_r, e_r)$ with continuously varying $r$. By discretizing $r$ as a uniform grid across the globe, there might be lots of values of $r$ for which the probability mass would be 0, and in regions of high probability, the discretization might hurt the granularity of the approximation. Modeling the problem with continuously varying $r$ seems more natural and yet is not considered in this work.

**(W3)**: Lack of presentation clarity. Portions of the paper are not well explained. The loss section particularly (3.4) is not explicitly tied to the variables and terms introduced in the problem definition section. As far as I can see, the task description for “species range prediction” is not explicitly defined (what is the input, the output etc.).

**(W4)**: Unclear experimental evaluation. It seems that the authors have compiled their own training and testing datasets. Why weren’t comparisons made directly on the datasets used for SINR (or other baselines) for example? Also, it looks like SINR has better geo-feature regression which suggests better geospatial representation learning, which is not discussed in detail in the experimental evaluation.

Overall, I think that the weaknesses of the paper outweigh its benefits and so I lean towards rejection. The proposed method is not especially novel (eg: the species embeddings are extracted from a pre-trained Llama 2 via natural language queries) and while a new metric is introduced, it is not thoroughly justified. I believe the authors could improve their work by being more precise in their presentation.

**Questions:**

What exactly is $\hat{r}$ in lines 223-224 and 234-235? A location embedding?

Have the authors tried llama 3 for the embeddings? Or maybe CLIP for better visual alignment?

---

### Meta-Review · Area_Chair_RRTb · 2024-12-17

**Metareview:**

This submission received four reviews, with reviewers expressing concerns about the validity of several assumptions and the absence of key comparisons. As the authors did not provide a rebuttal to address these points, the reviewers maintained their initial negative scores. We encourage the authors to carefully incorporate the feedback received to strengthen their work for future submissions.

**Additional Comments On Reviewer Discussion:**

There were no discussions since there were no response from authors.

---

### Decision · Program_Chairs · 2025-01-22

Reject